# Foraging Wild Edibles: Dietary Diversity in Expanded Food Systems

**DOI:** 10.3390/nu15214630

**Published:** 2023-10-31

**Authors:** Anne C. Bellows, Sudha Raj, Ellen Pitstick, Matthew R. Potteiger, Stewart A. W. Diemont

**Affiliations:** 1Department of Nutrition and Food Studies, Falk College, Syracuse University, Syracuse, NY 13244, USA; sraj@syr.edu (S.R.); pitstickellen@gmail.com (E.P.); 2Department of Landscape Architecture, College of Environmental Science and Forestry, State University of New York, Syracuse, NY 13210, USA; mpotteig@syr.edu; 3Department of Environmental Biology, College of Environmental Science and Forestry, State University of New York, Syracuse, NY 13210, USA; sdiemont@esf.edu

**Keywords:** food systems, diet diversity, foraging, foraging literacy, urban food forests, urban gardens, bioactive compounds, antioxidants, traditional ecological knowledge, nutrition transition, nutritional composition, chronic disease

## Abstract

Human food foraging in community forests offers extensive and expandable sources of food and high-quality nutrition that support chronic disease prevention and management and are underrepresented in US diets. Despite severe gaps in non-commercial “wild food” data, research in Syracuse, NY, identified substantial amounts of five key antioxidant phytochemicals in locally available, forageable foods with the potential to augment local dietary diversity and quality. Findings endorse the need for micro- and macro-nutrient research on an expanded range of forageable foods, community nutrition education on those foods, an expanded study on antioxidant phytochemical function, and the inclusion of forageables in the food system definition.

## 1. Introduction

This paper makes the case that human food foraging in community forests encompasses extensive and expandable sources of food and high-quality nutrition, notably antioxidant phytochemicals, that are underrepresented in US diets. Current US food consumption patterns misalign with the Dietary Guidelines Advisory Committee counsel that promotes foods with protective roles in decreasing the risk of all-cause mortality and chronic disease risk [1] and additionally supports environmental sustainability [2,3]. Instead of nutrient-dense and relatively unprocessed foods spread across six core food groups, i.e., vegetables, fruits, grains, dairy, proteins, and oils [4], residents consume few fruits, vegetables, whole grains, and dairy, and too many refined grains and foods containing added sugars, saturated fats, or high levels of sodium [5]. To counter the contribution of our globalized and commodified food system on present day diets [6,7], alternative forms of high-quality, more localized, and low- or no-cost food access are called for [8,9]. The study joins two discrete research questions: (a) What forageable plants, “wild” and intentionally introduced and grown without cultivation practices for commercial or domestic use, are available in Syracuse, NY, USA? (b) What is their nutrient content, including the presence of bioactive compounds, specifically antioxidant phytochemicals? The findings present arguments that foraging supports a largely untapped capacity to augment a community’s affordable, nutrient-dense dietary alternatives.

Foraging is a biocultural practice of nurture and exchange between humans and their local environment [10,11] that bolsters community nutrition and well-being across the world [12,13,14]. Shackleton et al. [15] introduce foraging in cities and towns, which is performed across undeveloped and built environments alike, as “the practice of harvesting or gathering raw biological resources within urban and peri-urban settings primarily for direct consumption, decoration, crafts, barter, or small-scale sale” (p. 2). Bukowski and Munsell [16] describe the “big picture” of urban food forestry as

“The intentional use of perennial food-producing plants to improve the sustainability and resilience of urban communities. A wide range of activities and projects fit this definition, including community food forests, urban orchards, foraging in public urban forests, and community-wide gleaning programs. Even maintaining a small food forest on a private residential lot in a city is considered urban food forestry” (p. 6).

Plants provide humans with macro- and micronutrients that are essential for optimal health and wellness by contributing to nutrient density and dietary diversity in mindfully planned diets. Additionally, plants produce phytochemicals that are bioactive nutrient and non-nutrient compounds that are able to attenuate metabolic imbalances such as oxidative stress and chronic inflammation in a protective capacity [17]. 

These metabolic imbalances are major contributors to chronic degenerative diseases such as cancer, metabolic syndrome, and cardiovascular disease [18,19]. Conventional allopathic medicine utilizes a variety of pharmaceuticals to treat these diseases, often with limited success and co-existing with the impact of side effects and questions related to the safety of such products. Phytochemicals such as polyphenols, carotenoids, anthocyanins, alkaloids, glycosides, saponins, and terpenes, as well as dietary antioxidants such as vitamins A, C and E, from natural whole foods such as fruits, vegetables, and spices, are dietary sources that have therapeutic potential to lessen the oxidative stress and inflammation associated with degenerative diseases. Dietary antioxidants such as vitamins A, C and E and other nutrients such as selenium and riboflavin that are essential cofactors for antioxidant enzymes participate in the process of scavenging free radicals, thereby reducing their degenerative potential in the chronic disease progression [20,21]. The complexities of their biological action remain a field of significant scientific interest. This paper focuses on a selected subset of antioxidant phytochemicals to be identified in forageable plants (such as lambsquarters, serviceberry, and black walnut), for which the available epidemiological evidence verifies their synergistic value in offering protection against chronic disease. In other words, the full, and only partially understood, power of antioxidant phytochemicals can be realized through foraged food sources. These bioactives, independently and in certain complementary combinations (or “packages”), can offer additive and synergistic benefits that may be more than the sum of individual benefits [22]. It is, however, important to recognize that (a) the action of these phytochemical bioactives is influenced by individual nutritional status, genetic predispositions, and socio-cultural and ecological environments and that (b) not all complex “packages” offer benefits and that certain bioactives and their associated complexes may also be antagonistic and not beneficial [23]. Phytochemicals such as saponins, terpenes, glycoalkaloids, etc., are categorized as antinutrients because they can bind to minerals, compromising their bioavailability. At low concentrations, antinutrients such as phytates, lectins and tannins can act as antioxidants and glucose and insulin modulators [24]. Fermentation, germination, milling, autoclaving, and soaking are food preparation techniques that can reduce the level of antinutrients in food [25]. 

The focus of this paper is on nutrient content, not toxicity risk. Authors readily acknowledge that addressing toxicity is as vital to the discussion of wild edibles as it is to commercially available foods. Concerns about contaminants in wild and commercial foods are not new and have been addressed with calls for caution in both cases [26]. In wild foods, these concerns range from the consumption of poisonous or allergy-inducing plants or parts of plants to risks associated with the potential presence of pesticides, herbicides, heavy metals, and polycyclic aromatic hydrocarbons (PAHs), among others [27,28,29,30,31,32,33]. Broad community education is of course required. Risk-reduction strategies can help, e.g., in avoiding difficult-to-identify plants (note that mushroom species are omitted from this article’s review) or urging newcomers to wild edibles to learn from experienced foragers in the field. Foraging education must be extended to local governments so that appropriate environmental policy can adapt to public food harvests [32,34]. For example, herbicide use must be curtailed or warning signs posted in a manner similar to how many municipalities provide signage and education on risks associated with urban fishing. A closer review of toxic risks is outside the scope of this paper.

This article derives from inquiry generated by the Syracuse Urban Food Forest Project, started in 2019 with goals of identifying existing forageables in an urban riparian corridor and augmenting them with supplemental plantings. Three steps shape our objective of demonstrating the antioxidant phytochemical potential in local, accessible, and low- or no-cost forageable foods. The first step is to identify good sources of the selected antioxidant phytochemicals (vitamins A, C, and E, selenium, and riboflavin) in forageable foods growing in the urban forests in Syracuse, NY. Where data are available, we report all found forageables and their phytochemical availability. The second step is to introduce a method to characterize an antioxidant “package” consisting of the five selected phytochemicals reviewed with the intention of portraying synergistic interactions of co-existing nutrient and non-nutrient bioactives. This method is applied to forageables for which relevant data are available for all five selected phytochemicals, as well as to select commercial plants. The final step is to expose gaps in available data on antioxidant phytochemicals in wild, forageable foods.

## 2. Antioxidants and Health

### 2.1. Diet-Related Chronic Disease and Food Supply

Nutrition-related non-communicable chronic diseases are recognized as the most significant contributors to morbidity and mortality in the world today. More than 50% of all deaths can be attributed to degenerative, inflammation-related diseases such as ischemic heart disease, stroke, cancer, diabetes mellitus, chronic kidney disease, non-alcoholic fatty liver disease, and autoimmune and neurodegenerative conditions [35]. Altered body composition associated with pathophysiological and/or metabolic anomalies such as oxidative stress and chronic inflammation is one of the well-documented risk factors at the nexus of the global chronic disease epidemic. A potent contributor to this growing epidemic is “nutrition transition”, a global-scale phenomenon marked by far-reaching alterations in the food supply. Across the world, food systems have been moving away from culturally based, traditional, and indigenous food economies grounded at the local and subsistence scale and moving toward cash crop production. Industrialized economies are characterized by consolidation of food production, economies of scale, money, and cash markets that determine what is produced and sold in the global food system. Selective breeding techniques, fortification and enrichment strategies, the expansion of functional foods and nutraceuticals, novel food processing and packaging technologies, and information and communication technologies have increased agricultural commodities’ yields, encouraged product innovation, diversification, and the volume of production of processed and ultraprocessed foods [36]. Concomitant advancements in the retail, market, advertising, and transportation sectors further ensure guaranteed availability of such products to global populations worldwide. However, much heterogeneity exists within and between global countries on the consequences of this industrialization and globalization depending on the ecological, internationalization, political, and technological climates that prevail within them. As local food markets have expanded to include more industrialized and commodified food products, traditional indigenous foods have been replaced by ultraprocessed foods high in sugars, processed fats, and sodium [37,38,39]. Cultural foodways, dietary traditions, and related food practices have been altered by pervasive industrialization and urbanization. These structural conditions behind nutrition transition correlate with uneven socio-economic improvement, wealth–poverty polarization, and attendant purchasing power disparities, as well as injury to cultural landscape ecologies. Habitat destruction, loss of land tenure, and rural out-migration have undermined traditional food activities like foraging. 

Under a nutrition transition, consumption is facilitated by the easy and often year-round accessibility of shelf-stable processed foods. At the same time, access to whole and nutrient-dense fruits, vegetables, and whole grains is limited for many segments of the population due to higher prices and sporadic availability. Further, traditional and Indigenous food access (e.g., food production for local markets, subsistence production, and foraging) and related cultural knowledge continue to decline, adding to the accessibility dilemma. Wild foraging has the capacity to provide consumers with nutritional variety, accessibility, and expanded food choice within their local food system. Altered dietary patterns along with concurrent lifestyle behaviors such as growing sedentarism have contributed to body composition changes, seen as overweight and obesity. Depending upon the degree and pace of these modifications, different parts of the globe have experienced a mix of positive and negative consequences. The net result, however, is population-scale conversion to poorer-quality diets characterized by more processed products that are rich in undesirable ingredients such as fats, oils, and sweeteners and often lacking nutrient-dense foods that can offer protection against chronic diseases [18,40]. 

At the global scale, we additionally observe a shift from undernutrition as the main nutrition problem to a triple burden of disease. The latter is characterized by overnutrition and undernutrition coupled with maternal mortality that is reflective of extensive population-scale food and nutrition insecurity, nutritional deficiency and excess in some cases, and the persisting burden of infectious diseases. Although distinct in terms of caloric excess, nutritional well-being has become increasingly limited or extinct in overnutrition cases. Overall, poor-quality diets marked by calorie- and energy-dense foods continue to replace nutrient-dense options, resulting in a dietary landscape wherein chronic diseases such as diabetes, cardiovascular disease, cancer, etc., prevail. Additionally, the poor micronutrient status of these diets affords little immunity against infectious diseases. Poverty, food insecurity, exposure to environmental toxins, and income and health access disparities further undermine the health of global populations. These factors, exacerbated by genetic predisposition and chronic stress, overburden the healthcare system’s capacity to simultaneously handle the dual burden of co-existing communicable and non-communicable diseases and lead to lowered immunity and chronic disease across the lifespan [41,42,43,44]. 

Researchers interested in the fetal origins of adult-onset disease view the triple burden of malnutrition in pregnancy and early fetal life as a major threat that increases the susceptibility to chronic metabolic diseases such as diabetes in later life [43,44]. Poor nutritional status accompanied by metabolic disturbances such as insulin resistance in pregnancy mediated by chronic inflammation and oxidative stress leads to a compromised placental nutrient supply to the fetus [45]. This can lead to developmental adaptations resulting in a compromised or altered organ structure, for instance, a pancreas with fewer islet cells and altered functional capacity accompanied by altered metabolic, endocrine, nutrient sensing and inflammatory signaling pathways [43,45,46,47,48]. 

### 2.2. Chronic Disease and the Underlying Dysfunctions in Oxidative Stress, Inflammation, and Detoxification

Current research suggests that chronic disease occurs when three inherent and necessary metabolic processes—namely oxidative stress, inflammation, and detoxification—in the human body become less resilient [41,49]. These processes, independently and synergistically, are integral for humans to function. For example, the acute inflammatory response that is critical to recovering from trauma and oxidative stress produces free radicals for pathogen destruction. However, these processes can become vicious and self-perpetuating under abnormal circumstances, such as stress impacting various physiological systems and ultimately causing disease. Restoring balance involves strengthening the physiological systems to better handle these processes by addressing genetic anomalies and the provision of good-quality nutrition, living environments, and positive lifestyle behaviors.

Oxidative stress occurs when there is an imbalance between the production of highly unstable free radicals such as reactive oxygen and nitrogen species with an unpaired electron and the body’s antioxidant defenses to address these free radicals [50,51,52]. Free radicals with one or more unpaired electrons are normally created in the body via a variety of physiological processes (e.g., respiration) and genetic abnormalities (e.g., inborn errors of metabolism) and as a result of exposure (e.g., pollution). Within limits, free radicals are critical in cellular processes such as cell signaling. Under normal physiological conditions, free radicals are dealt with by a combination of endogenous antioxidant systems produced within the body such as enzymes like catalase, superoxide dismutase, and exogenous antioxidants such as vitamins C, A, and E obtained from the diet. These systems individually and in combination neutralize the free radicals by giving up their own electrons and maintaining a steady supply of available antioxidants through the processes of oxidation and reduction [31,51,52]. However, adverse chronic environmental exposures and dysfunctional physiological processes can lead to a buildup of free radicals, resulting in oxidative stress. This situation is further exacerbated by an inadequate intake of antioxidants from food and/or insufficient or inefficient functioning of endogenous antioxidant systems. These condition results in both the consequent inhibition of the body’s protective defensive mechanisms and damage from free radicals. The uncontrolled production of free radicals can result in, for example, damage to the lipid-rich cellular membranes, the creation of nonfunctional proteins, and DNA damage causing mutagenesis [51]. This occurs because free radicals indiscriminately target biomolecules like lipids, proteins, and DNA in the hopes of securing their missing electron. 

Inflammation (acute and chronic) is a normal protective response that is critical to survival yet when unabated can become a threat to human health. The acute inflammatory response is seen in conditions characterized by tissue damage due to trauma, microbial invasion, or noxious compounds, as is the case when one gets a bruise or when a cut becomes infected. Under these circumstances, several immune system components participate in eliciting the inflammatory response. Acute inflammatory responses are typically resolved to baseline levels in a short period of time depending on the efficiency and functionality of the immune system. In chronic inflammation, the immune response is initiated in a similar manner, but the degree of resolution is modulated by genetic predispositions or environmental factors such as diet, physical inactivity, chronic stress, and lifestyle behaviors. A persistent hyperactivated stress response and/or a dysfunctional immune response accompanied by undesirable nutrition and lifestyle factors make the resolution of inflammation unachievable. Chronic inflammation is marked by increasing levels of pro-inflammatory mediators such as Interleukin-6 and TNF-alpha, cortisol buildup, and insulin resistance that eventually culminate in disease. The intensity and impact of the inflammatory response appear to be highly individualized [53]. The effective resolution of the inflammatory response is regulated by an arsenal of endogenous and dietary anti-inflammatory mediators. These mediators can synergistically and effectively inhibit or block actions that perpetuate inflammation, thereby preventing the progression of the chronic inflammatory state [54]. 

Detoxification is the third process by which the body identifies, neutralizes, and eliminates toxic substances. This process is also influenced by lifestyle and environmental factors, the total toxic load experienced by an individual determined by exposure to toxins, the efficiency of the body’s endogenous detoxification enzyme systems that involve the Phase I and Phase II cytochrome P450 enzyme systems, and the availability of nutrient and non-nutrient cofactors for the proper action of these enzyme systems [55,56].

### 2.3. Reducing Oxidative Stress and Chronic Inflammation

Dietary patterns rich in minimally processed, nutrient-dense foods and low in ultra-processed foods are associated with a reduced risk of unimpeded oxidative stress, chronic inflammation, and enhanced detoxification processes. Whole foods such as fruits, vegetables, whole grains, nuts, legumes, etc., contain a plethora of bioactive compounds that include antioxidant phytochemicals, macro- and micronutrients such as lipid bioactives, vitamins, minerals, and dietary fibers in natural ratios and proportions [57]. These compounds, when present in their innate food matrices, in contrast to industrially created dietary supplements, act in an independent, complementary, or synergistic manner to maximize disease prevention and offer a therapeutic potential beyond nutrition [58]. Food bioactives exert their therapeutic and protective potential in several ways. These include influencing cell signaling pathways, engaging in scavenging free radicals, interacting with transcription factors such as NF-kB and Nrf2 that are related to anti-inflammatory and antioxidant effects, as well as enhancing anti-inflammatory effects via the production of short-chain fatty acids by the gut microbiota. In this regard, whole foods, with their innate combinations of bioactive antioxidants and phytochemicals, can keep processes such as oxidative stress and chronic inflammation at bay. Although several thousand food bioactives have been identified, much remains to be learned about their bioavailability, mode of action, and dietary requirements [59].

There is no simple guide to an ideal composition of antioxidant phytochemicals that can ameliorate unimpeded oxidative stress and chronic inflammation and that can boost detoxification processes [60,61,62]. Yet there is substantial scientific evidence that indicates higher dietary intakes and/or blood concentrations of vitamin C, carotenoids and tocopherol are associated with reduced risk of cardiovascular disease, cancer and total mortality [63]. Researchers further highlight that these benefits are accrued not due to the individual antioxidants themselves but the combination of the various bioactive compounds that coexist within the food matrix, underscoring the importance of consistently higher dietary intakes of a diverse assortment of whole foods. This recommendation is made based on the premise that there is a constantly fluctuating balance between the proinflammatory and oxidative processes that occur in the body and the available exogenous dietary antioxidants and endogenous antioxidant enzyme systems to handle these processes. Additionally, while genetics determine the endogenous antioxidant profile, the availability of exogenous antioxidants varies with individual dietary patterns, dietary diversity, seasonality, the local availability and accessibility of whole foods, and appropriate behavior-modification efforts. This underscores the importance of increasing consumer awareness of the value of such products and encouraging their use in chronic disease prevention and management [64].

The argument is made that dietary patterns such as the Mediterranean diet featuring a predominance of whole foods provide a food bioactive package that offers significant health benefits and immune-boosting capabilities [65]. A steady supply of these bioactive compounds in an appropriately planned meal pattern can assist in overcoming the imbalance created by the free radicals and proinflammatory compounds. Through a variety of mechanisms such as anti-inflammatory and chemo-preventive effects, food bioactives can simultaneously provide various benefits that can offer additive and synergistic advantages [58]. A beneficial antioxidant bioactive “package” would then include both “direct antioxidants” (like vitamins A, C, and E that themselves participate in free radical scavenging) and “helper antioxidants” (like riboflavin and selenium that participate as cofactors for important anti-inflammatory endogenous enzyme systems such as glutathione peroxidase) [66,67], as well as efficiently support detoxification enzyme systems such as the Phase 1 and 2 enzymes [68,69].

## 3. Method

### 3.1. Case Study Background

#### 3.1.1. Location

The Syracuse Urban Food Forest Project (SUFFP) aims to (a) learn how edible urban forests can be structured to optimize human and wildlife health and well-being; (b) work with communities to share and expand knowledge about the social benefits of public harvests in urban forests; and (c) recognize and respond to historical and contemporary infringements on rights of land tenure, public land access, and green space equity. To examine whether a local, forageable food system can enhance dietary diversity, we chose to examine total antioxidant and phytochemical availability as a first step. This strategy could reveal local foods that can potentially reduce chronic disease risk in the population. The SUFFP, initiated in 2019, focuses on a seven-mile corridor on the south and southwest side of Syracuse, NY. The riparian transect runs along Furnace Brook to its intersection with Onondaga Creek and northward through the city center to Lake Onondaga. The corridor is surrounded by public land, parks, schools, residential low-low-income and resurgent (identified by the city for needed improvements) neighborhoods, and two urban farms (see Figure 1) [70].

#### 3.1.2. Poverty, Urban Food Infrastructure, and Health Nexus

Southside Syracuse experiences social and economic isolation and inequality that, in turn, precipitate poor health and nutrition outcomes. The land sits on unceded Onondaga Territory, but the actual number of tribal members of this branch of the Haudenosaunee living within the Syracuse city limits is quite small. As of 2020, the overall population of Syracuse was 148,620 [71], with an estimated 30.3% of Syracuse residents living below the poverty level (more recent US census data are available but do not include the American Indian and Alaskan Native population due to limited sample size) [72]. The city’s history of colonization, redlining, and environmental racism has had lasting effects visible through segregated poverty levels, where 33.3% of American Indian and Alaskan Native, 40.0% of Black or African American, and 42.9% of the Hispanic or Latino populations live below the poverty level, compared to only 23.3% of the White population [72,73]. In 2020, the estimated median income in Syracuse was USD 38,893. American Indian and Alaskan Native residents were estimated to earn a median income of USD 23,316, Black residents a median income of USD 30,142, and Hispanic or Latino residents a median income of USD 33,310, while the median White income was estimated at USD 46,177 [72]. The Southside neighborhood of Syracuse, where our study takes place, is predominantly home to people of color, particularly Black or African American residents [71]. The 2020 Syracuse Urban Forest Master Plan acknowledges greenspace alignments with higher income and more white neighborhoods and defines strategies and sets goals of “canopy equity” for City’s Division of Forestry [74].

Income and food access infrastructure influence food and nutrition security and, in turn, susceptibility to chronic disease [75]. The prevalence of chronic disease throughout Syracuse varies along neighborhood lines [76]. Model-based estimates of chronic disease rates across the United States were calculated via census tract as part of the 2019 Behavioral Risk Factor Surveillance System (BRFSS). In the census tracts along the SUFFP transect, chronic disease rates are generally higher than the City of Syracuse and the national average (Table 1). Reports also note limited availability of high-quality and full-service markets on the Southside or easy public transportation routes to access such food outside of the area [77]. Of the four census tracts that intersect with the Southside neighborhood (New York tracts 42, 53, 54, and 58), three are classified by the USDA as being food-insecure (Table 1). In other words, populations along the SUFFP transect experience high or higher-than-average rates of diet-related chronic disease, live in low-income households, and have limited access to healthy food needed to address related nutrition deficiencies.

Correlation of poverty, lack of healthy food access, and chronic disease incidence is known [75]. While we have not conclusively established these connections on Syracuse’s Southside, available data indicate that the availability of whole and healthy food alternatives would be welcome.

#### 3.1.3. Forageable Species Selection and Antioxidant Phytochemical Data Collection

Syracuse is in the US Department of Agriculture Plant Hardiness Zone 5b, reflected by lowest annual temperatures of −15 to −10°F [79]. Under a US Forest Service, Great Lakes Restoration Initiative (GLRI) grant, SUFFP is expanding edible tree, shrub, and other ground foliage on public land, including parks and vacant lots. The SUFFP project objectives include modeling the impact of new planting projects on stormwater run-off and nutrition availability.

In 2019 and 2020, field testing in 12 unique sampling areas of upland forest, upland grassland/field, wetland, and riparian eco-zones along the seven-mile transect identified 196 native and non-native edible species growing “wild”, i.e., without standard practices for commercial cultivation or kitchen gardens. In 2020 and 2021, with the intention of a future “wild” edible planting effort, SUFFP assembled a list of plants to augment and complement the available edible species for its GLRI proposal. Plant selection attempted to embrace flora that the local population might already know and recognize, as well as less familiar ones. 

In the literature above, plant-based sources of nutrition from “whole food” (i.e., less processed or unprocessed food) are identified for their potential to reduce, synergistically, the risk of chronic disease because of their high sources of antioxidant phytochemicals and the corresponding immune system and anti-inflammatory function. The literature also underscores that there is no perfect “package” of antioxidants, but that the ideal complement varies according to individual and environmental factors that are in constant fluctuation.

#### 3.1.4. Antioxidant Phytochemical Data Collection

For this paper, we focused only on those plants found within the SUFFP transect for which we could find data on a combination of select direct and helper antioxidants (vitamins A, C, and E, selenium, and/or riboflavin) content. Of the 196 identified species, 38 species were selected and organized into five groups: fruits, herbs, nuts, vegetables, and other. For selection, ready availability in the food system, established evidence of effectiveness (as defined in the section above), and a manageable number were considered for this research paper. The primary source for species’ nutrient compositions was the USDA FoodData Central (FDC) database [80]. Strategies to locate missing data points included referrals to agricultural extension offices and outreach to local ethnobotanists and nutrition scientists (see Appendix A Table A1 for a full list and data sources). It is entirely possible that isolated data exist that researchers have not been able to locate. The authors welcome contact with further leads.

Commonly appearing vitamins, A, C, and E, as direct antioxidants, serve as radical scavengers interrupting oxidative stress processes [66,67]. As helper antioxidants, micronutrients such as selenium [68] and riboflavin [69] work as cofactors and co-enzymes for endogenous antioxidant enzymes such as thioredoxin reductase and glutathione peroxidase, respectively, that are involved in free radical scavenging, anti-inflammatory, and immune-boosting processes. Although selenium and riboflavin are not necessarily considered antioxidants, they are important because the activity of endogenous antioxidant enzymes cannot ensue without them. 

Cases where possibly comparable food values were not applied. We did not use available antioxidant values for one species of a plant for other related species. For example, we did not apply the butternut values (*Juglans cinerea*) to heartnut (*J. ailantifolia*). We could also have used known commercial values for a comparable but different plant. For example, we could have used FDC values for pecan (*Carya illinoinensis*) to evaluate other hickories, including bitternut (*C. cordiformis*), pignut (*C. glabra*), shellbark (*C. laciniosa*), and shagbark (*C. ovata*). This project has not adopted that course for several reasons: to emphasize the need for more related nutrition research on wild edibles; the fact that wild edibles differ in unpredictable ways; and in part because the assigned score method discussed above already collapses some level of data.

Cases where possibly comparable food values were applied. Commercially available fruits, vegetables, nuts, and medicinals comprise both native and non-native species, all of which have been genetically altered over time to address size, productivity, taste, etc. Forageable plants in both urban and rural Central New York, as elsewhere, include both native and introduced non-native species, the latter of which might be found in the form of abandoned orchards or invasive but edible undergrowth like Japanese knotweed (*Polygonum cuspidatum*). For this reason, we have used FDC commercial plant values for forageables like apples (*Malus domestica* used for *Malus* spp.), pears (*Pyrus communis* used for *Pyrus* spp.), and watercress (*Nasturtium officinale).*

Comparable food values were also applied to those foods included in FDC data that lack specific species identification. For example, we used the FDC data for non-specific hazelnuts (*Corylus* spp.) to evaluate American hazelnuts (*Corylus americana*). This was also performed with gooseberry (*Ribes* spp. used for *Ribes uva-crispa*), blackberry (*Rubus* spp. used for *Rubus fruticosus*), and black elderberry (*Sambucus* spp. used for *Sambucus canadensis*).

### 3.2. Case Study Analysis 

This paper next presents three stages of analysis on the antioxidant potential in the local forageable environment of southside Syracuse. First, we demonstrate the relative power of distinct antioxidants (vitamins A, C, E, selenium, and riboflavin) in forageable species along the Syracuse riparian corridor described above. In Stage 2, we introduce these species in terms of “antioxidant packages” by grouping the distinct antioxidants together (vitamins A + C + E + selenium + riboflavin), ranking the “packages” within organized categories of fruits, herbs, nuts, and vegetables. In Stage 3, we depict the forageable plant foods with the strongest-ranked (or one of the strongest, where equal) “antioxidant package” in each category (fruits, herbs, nuts, vegetables) with the most popular commercial food in each category.

#### 3.2.1. Stage 1 Analysis: Antioxidant Phytochemical Concentration and FDA Daily Value by Individual Plant Species

Plant concentrations per 100 g of vitamins A, C, E, selenium, and riboflavin are re-interpreted through the National Institutes of Health’s (NIH) established recommended allowances for macro- and micronutrients. Recommended Dietary Allowances (RDAs) are designed to guide individuals in planning nutritionally adequate diets. RDAs vary according to age, sex, and pregnancy or lactation status [81]. To simplify public understanding, however, the US Food and Drug Administration (FDA) developed a single Daily Value (DV) per nutrient. It is noteworthy that the DV is based on the RDAs for males aged 19 or older [82], thereby possibly eliding specific needs of females, young males, females in various reproductive stages, and all older persons. 

The FDA has also developed a system to identify diverse foods as being high, medium, or low (or no) nutrient sources of individual macro- and micronutrients in terms of their percent DV. As seen in Table 2, a food is a high source of a nutrient if it provides 20% or more of the DV per serving. Medium sources provide 5–20% of a nutrient’s DV per serving. Low sources contain 5% or less of the nutrient DV per serving [83]. “No value” is for foods with zero evidence of the nutrient in the food. Table 2 identifies the high, medium, and low threshold values for vitamins A, C, E, riboflavin, and selenium. 

The authors further created assigned scores (0–3) that correspond to the FDA’s high, medium, and low sources for vitamins A, C, and E and selenium and riboflavin. Table 2 displays the DVs for vitamins A, C, E, selenium, and riboflavin and correlates the assigned scores with the FDA nutrient source threshold values. A value of “3” reflects a high nutrient source; “2”, a medium source; and “1”, a low source. Zero (0) describes a dearth of nutrient availability. 

In the paper’s Results section, Table 3 reports antioxidant phytochemical values per 100 g of those species for which there are known antioxidant values. Additionally, the tables introduce the antioxidant concentration in terms of high, medium, low, or no concentration value (per FDA protocol) as well as an assigned score (3, 2, 1, 0, respectively, as outlined in Table 2). Plant species are organized from highest to lowest antioxidant concentration per 100 g. 

#### 3.2.2. Stage 2 Analysis: Plants Ranked by Power of Composite (or “Packaged”) Antioxidant Scores

In this stage of analysis, antioxidant scores (3, 2, 1, 0) are added together for all forageable plants for which all five reviewed antioxidants were known (Table 4). The resulting composite assigned score reveals the relative potential of diverse plants in terms of the number and power of their constituent complementary antioxidants. Table 4 groups these plants according to type (fruits, herbs, nuts, vegetables, and other) and ranks them within their type using a composite assigned score power. 

#### 3.2.3. Stage 3 Analysis: Reflections on Antioxidant Phytochemical Packages of High-Value Forageables and Popular Commercial Plant Foods according to Food Type (Fruits, Herbs, Nuts, and Vegetables)

To assess the contribution of forageable foods to antioxidant phytochemical consumption, Table 5 presents forageable and commercial foods side-by-side according to food type (fruits, herbs, nuts, and vegetables) using (a) the forageable food with the strongest (or one of the strongest, where equal) ranked “antioxidant package” in each food type and (b) the most popular commercial food in the same category. There is one exception. In the vegetables category, we include romaine lettuce (the most popular leafy green) instead of onion (the most popular vegetable overall) to be better portrayed alongside the forageable dandelion greens. The visualization of Table 5 data is further enhanced through the inclusion of radar charts (Figure 2 and Figure 3).

#### 3.2.4. Limitations

The most critical limitation has been the lack of available antioxidant data for forageables. To the extent that nutrient data on wild edibles are at all extant, they are restricted to those plants comparable to commercial products (e.g., black walnuts or mulberries) and occasionally consumed in contemporary diets; they are not reflective of the broad spectrum of plants central to traditional and Indigenous diets.

We further emphasize that available data (e.g., Table 3) have not been validated under Syracuse conditions. Plants adapt to local conditions, affecting nutrient values that can vary according to soil type and quality, USDA plant hardiness zone, and the vast diversity of plant types known as cultivars, among other factors. Selenium (Se) in plant-based foods provides a good example, as availability varies according to the plant’s accumulation ability and the Se content of soils [84,85]. The United States is a Se-rich region with the Se content of most soil samples ranging from 0.1 to 2 mg/Kg and never exceeding 100 mg/kg [84,86]. The mean Se value in Onondaga County, where Syracuse is located, is 0.277 mg/kg [87]. FDC values do not reflect nutrient values with this level of plant specificity. Thus, available antioxidant values must be viewed as best estimates until more regionally and cultivar-specific data can be obtained.

An additional limitation concerns human nutrition impact assessments according to presumed plant quantities consumed. Nutrient availability is notably connected to quantities eaten, which is greatly determined in forageable and commercial plant foods by seasonality, cost, familiarity, accessibility, and preparation methods. Nutrient assessment does not, of course, lead to or predict access and incorporation into the diet. Like nutritious commercial foods, just because a nutritious “wild” food is available does not mean people recognize it, have the means to obtain it, know how to prepare it, or want to eat it. A discussion of the actual incorporation of forageable plants into the diet is outside the scope of this paper but certainly a topic for future investigation. We note that such a discussion must address policy and planning to expand and protect plant availability and safety, as well as to promote related knowledge and training to nutrition educators and diverse food programs. It requires addressing local foraging knowledge (including understanding of the ethics of foraging—see, e.g., Kimmerer 2013 [10]) and rudimentary plant-processing skills. Foraging for auxiliary nutrient support also presumes peoples’ awareness, time, interest, cultural food inclinations and preferences and the availability of foraging and cooking tools, among other factors which must be considered.

## 4. Results

The Results section reviews vitamins A, C, and E, selenium, and riboflavin concentrations in the edible portions of forageables found growing “wild” or planted as part of the SUFFP transect (Table 3). Table 4 presents composite scores for five nutrients according to whole forageable foods grouped according to fruits, herbs, nuts, vegetables, and others. Table 5 displays the highest-ranking composite scores for forageable foods according to group against the composite score for the most popular commercially available food according to group [88,89]. The radar charts that follow assist with visualizing the nutrient potential of forageables.

In Table 3, vitamin A concentrations (in mcg per 100 g) show values ranging from 1 to 1380 mcg. The DV for vitamin A is 900 mcg, with high sources containing at least 180 mcg per serving. High plant-based sources of vitamin A include leafy green vegetables, orange and yellow vegetables, and tomato products [90]. Our data on forageables show high vitamin A values in grape leaves, lambsquarters, dandelion greens, ground sage, spearmint leaves, and ostrich fern. 

Vitamin C concentrations (in mg per 100 g) exhibit values ranging from 0.7 to 181 mg. The DV for vitamin C is 90 mg, with high sources of at least 18mg per serving and tending toward citrus fruits, potatoes, and tomato products [91]. Our data reveal high vitamin C content in blackcurrants, ramps, lambsquarters, quince, and persimmon, among many other vegetables, fruits, and herbs.

Foods naturally high in vitamin E include nuts, seeds, and leafy green vegetables [92]. The DV for vitamin E is 15 mg, with high sources containing at least 3 mg per serving. Our data on forageables (in mg per 100 g) range from 0.70 to 15 mg, with the highest sources identified as hazelnut, ground sage, and dandelion greens. 

Table 3 displays selenium concentrations (in mcg per 100 g) ranging from 0 to 18.3 mcg. The DV for selenium is 55 mcg; the high sources contain at least 11 mcg per serving. While foods naturally high in selenium include muscle meats, cereals, and dairy products [85], our data reveal high sources in hackberry, butternut, and black walnut.

Riboflavin concentrations (in mg per 100 g) range in value from 0.02 to 3.54 mg. The DV for riboflavin is 1.3 mg, with high sources containing at least 0.26 mg per serving. Foods naturally high in riboflavin tend toward meats, eggs, and milk, including few vegetables, with rare exceptions like spinach [93]. Nevertheless, our data show high riboflavin sources in Saskatoon serviceberry, lambsquarters, grape leaves, ground sage, and dandelion greens. 

As seen in Table 4 and based on the wild foods for which we have complete data, herbs and vegetables have stronger composite antioxidant phytochemical packages than other forageables. The “composite score” communicates strength pertaining to our selected package of antioxidant phytochemicals. 

Table 5 displays composite scores for one forageable and one commercial plant in each category (fruits, herbs, nuts, and vegetables). The forageable plants chosen had the highest composite score in each category. The commercial plants selected for the fruits and nuts categories represent the most popular respective commercial consumables in 2021 according to USDA per capita availability [88,89]. Note that the USDA ERS does not maintain data on herbs, so the authors selected basil as the commercial plant in the herb category based on their own understanding of the popularity of herbs in the US. To better illustrate the antioxidant potential of dandelion greens in the vegetables category, the most popular commercial leafy green, romaine lettuce, was selected over the most popular commercial vegetable overall, i.e., the onion. 

Table 5 shows that the selected forageable foods within each category have composite scores that are at least on par with (and sometimes better than) popular commercial foods for the same categories. This suggests that forageables can contribute successfully to dietary diversity through the consumption of varied and complementary antioxidant-rich foods.

The radar charts in Figure 2 are based on Table 5. Selected forageables and commercial foods provide a graphic illustration of the assessed composite antioxidant phytochemical profiles. Fuller profiles correspond with higher composite scores. Figure 3 presents the amalgamated antioxidant profiles of forageable and commercial foods, respectively. Measuring the synergistic (and possibly even antagonistic) interactions of these compound profiles is beyond the scope of this paper. Yet based on the literature cited earlier, one can expect that, when eaten together in whole or minimally processed form, complementary and positive antioxidant interactions associated with dietary diversity likely occur. 

Composites for the selected foraged foods exhibit analogous or more complete profiles than do the composites for commercially popular foods in the same food category (e.g., with fruits, 2—black mulberry vs. 3—strawberry; with herbs: 2—sage vs. 3—basil). Again, this provides evidence for the potential of forageable foods to contribute to overall dietary diversity, quality, and quantity. Actual accessibility of any or all the selected foods varies dramatically according to many factors, including seasonality, availability in the local outdoors or local supermarkets, and eaters’ familiarity and experience with the different foods. 

## 5. Discussion

Motivated by questions of foraging benefits in and around urban centers, this paper provides evidence of the nutritional value of antioxidant phytochemicals in existing and planned food forests in Syracuse, NY, USA. Our efforts led to an increased understanding of the value of locally available low- and no-cost high-quality dietary fortification against massive population-scale chronic disease prevalence (diabetes, heart disease, etc.). Piecing together this information revealed present and meaningful nutrient density in “wild” foods. 

The project also exposed enormous gaps in nutrition data for non-commercial wild forageable foods in the US. While research in other countries on wild edible plants is bountiful [29,94,95,96,97,98,99,100], efforts within the United States are severely limited and warrant future study. The lack of complete data in our study could have been adjusted through the insertion of nutrient values of comparable foods to broaden the available plant data. 

Globally, knowledge about nutritional diversity and native plant availability is embedded in local communities and their food systems. These knowledges of survival in, and co-existence with, the uncommercialized natural world is transmitted through generations. They value sustainable and symbiotic relationships between living beings in the environment, both human and non-human. Our findings, however, lead us to recognize and reflect on the lack of community knowledge of nutritional abundance in local food systems. “Modern” food systems rely on human relationships with their markets, undermining the cultural capital gained from reproducing time-tested nutritional well-being through intra-generational foraging practices.

### 5.1. Available Nutrition Data: Implications of What We Could Find

A variety of forageable plants in Syracuse (fruits, herbs, nuts, vegetables) contain the US FDA Daily Value recommendations of antioxidants A, C, E, selenium, and riboflavin at medium and high levels per 100 g of plant material. Leafy greens and herbs supply high values of vitamin A. Fruits and leafy greens provide high values of vitamin C. Select nuts, herbs, and leafy greens offer the highest concentrations of vitamin E. Some berries and nuts stand out for selenium content. A diversity of fruits, maple syrup, leafy greens, and herbs contain the highest riboflavin scores. This information unmistakably points to the existence of a diverse and untapped availability of nutrients in one region of a nation prone to poor diets and increasing incidence of chronic disease. Although improved diet quality can assist in addressing the underlying bodily dysfunction associated with the national chronic disease emergency, commercially available healthier, less processed, and whole foods cost much more than ultra-processed foods with negligible nutritional value [101]. This limits access to nutrient-dense whole foods by major population segments such as those residing in the Syracuse Southside community. 

As noted in the paper’s literature review section, research on antioxidant phytochemicals is an expanding frontier in nutrition science. As much as there is to learn about individual nutrients, even more may yet unfold in investigations of the synergies resulting from nutrient–nutrient as well as nutrient–non-nutrient interactions within food matrices. It is important to recognize that no plant food is made up of only one nutrient, but rather is a complex assembly of nutrients and non-nutrients that interact physically and chemically to exert a myriad of therapeutically beneficial effects [102,103]. From a nutrition perspective, the value and functionality of these bioactives are to a large extent dependent on multiple factors: (a) the form it is consumed in, that is, whole food or as an individual component within a supplement, e.g., a whole orange versus a vitamin C supplement; (b) bio-accessibility, that is, the release of nutrients from the matrix, e.g., the release of minerals such as selenium from fibrous components; (c) the bioavailability of the bioactive, e.g., the availability of the antioxidant vitamins to the systemic circulation for functionality in the body; (d) the bioconversion of the bioavailable nutrient into an active form for functionality, e.g., retinol from provitamin A; (e) the bioactivity or specific effects of a compound in the body, e.g., antioxidant effects; and (f) bio-efficacy, or the production of a desirable or undesirable effect (e.g., radical scavenging activity) by the antioxidant fraction of an ingested, activated bioactive compound [103]. Food bioactive contents within plant foods is influenced by genotype, environmental factors (e.g., soil type, sun exposure, rainfall, climate change), environmental stress (e.g., exposure to light, agronomic factors like organic versus conventional cultivation), and processing (e.g., pasteurization, bleaching, roasting) [104]. This bioactive content is further shaped by the eater’s genetics, heritable predispositions, and environmental contexts of eating [105] that will ultimately impinge on the bioavailability and functionality of these compounds within the eater’s body. These myriad challenges to bioactive contents complicate the delineation of dietary recommendations, as well the predictability of potential interactions between these components in whole foods. Nevertheless, it is reasonable to conclude that if people have access to and consume a diverse plate of nutrient-dense whole foods, several health benefits can be gained through the coordinated activities of their countless components in the long run [58].

In line with available data, we have shown which plants appear to be the best sources of a complement of the studied antioxidants. We acknowledge that plants contain many more antioxidants than the five selected for this study. We also point out that nutrient data vary according to local soils, local climate, and other factors; some of our data did not even originate in the United States (yarrow data, for example, originated in Portugal and Spain). In this study, the best conceptual antioxidant “composite package” was noted in black mulberry among fruits; sage among herbs; black walnut among nuts; and dandelion greens among vegetables. Further, the conceptual forageable plant antioxidant packages present at least as strongly as, or better than, their popular commercial counterparts (strawberry, basil, almond, and romaine lettuce). Forageables present a low-cost, easily accessible opportunity for whole foods with nutrient-dense potential. This is critical, especially given the poor nutritional status nationwide made worse by poverty conditions such as in Syracuse. To that end, public health and social justice-focused policies are called for to address (a) the counter-productive high costs of accessing nutrient-dense whole foods; (b) the expansion of the popular understanding of wild forageables and their contribution to dietary diversity; and (c) the local geographies of healthy food systems. Additionally, but beyond the scope of this paper, municipalities should consider the availability of community commons for secure and tenured publicly accessible food foraging spaces. 

### 5.2. Non-Available Nutrition Data: Implications of What We Could Not Find

Food exists as a human right and the common heritage of all people [106]. As cultural capital, food shapes the health, well-being, and identity of communities. So much indigenous and traditional food and ecological knowledge that sustained the socioeconomic, nutritional, and pharmacological needs of people for eons has been lost to colonization and market-dominated economies [107,108]. Across the globe, Indigenous and local subsistence food systems are marked by traditions transmitted across generations that are instrumental in supporting local food culture and foodways. Enduring local food systems conserve and reproduce native plant varieties and their wild relatives; they offer food diversity and nutritional and medical security to local human and non-human communities while concomitantly maintaining agricultural and genetic biodiversity and environmental sustainability [109,110]. The loss of such systems and knowledges that protect and employ them, endangers food and nutrition security, as well as medical security, and erodes the cultural identity of local communities [111,112,113]. Foraging is an alternate food access strategy that can help (re)build a community’s cultural food capital, in part through restoration of, and edification about, bio-cultural capital [114,115]. Bio-cultural capital arises from regionally and culturally specific knowledge of how people and their local ecology can co-exist and sustain each other [116,117,118]. Bio-cultural capital’s Indigenous foundation confronts and variously absorbs food capitals—from root stock to foraging practices—from migrant, immigrant, and refugee populations [119]. Cultural capital, rooted in food-based relationships with the land, benefits from sustainable food practices, as well as social relationships based on collaboration and trust [120]. Lyson famously wrote about the capacity of local food systems and alternative food exchanges like community-supported agriculture to foster well-being in civil society [121]. Foraging amplifies his concept of “civic agriculture” through its intertwined relationship with nature, its reach for historical and Indigenous knowledges, and its greater disentanglement from the mainstream economy. With ethical foraging and honorable harvests, one gathers with consideration of plant regeneration and other human and non-human needs [10]. These practices, especially on public land, provide not only shared nutrition but also space to exchange bio-cultural information about health, well-being, and building resilience via ensuring food for humans, the animal and plant ecological infrastructure that sustain our environment [16]. When foraging practice wanes, critical ecological and cultural knowledge weakens, and community health becomes compromised.

We question why foods once central to the population’s dietary balance and health are not represented in popular nutrition data access [122]. Our efforts to provide antioxidant phytochemical data as a strategy to showcase the dietary value of “wild” foods play into a modernist mind frame that privileges technical evidence-based knowledge (EBK) over traditional ecological knowledge (TEK) [10,123]. Cultural and Indigenous food traditions nurtured centuries of humans before EBK identified the nutritional profiles of mac ‘n’ cheese and fruit punch. In such an environment TEK struggles for recognition in public policy. EBK data are expensive; the more informal and sometimes oral traditions of TEK cost less, little, or nothing. EBK data have been used largely for research oriented to the commodity scale, nutritional fortification, highly processed food, long shelf life, and the agro-business [124,125,126]. Foraging does not serve the marketplace. EBK and its presumed authority assign value to hundreds of chocolate bar varieties because of the profit margin, but not to the mayapples and chinkapin oak that are not sold in big box stores. The dominance of EBK overwhelms public confidence in TEK. It drives doubt into cultural knowledge and heritage, and it disturbs the sustainable ecological relationship that cultures build with TEK across time.

In presenting our findings, we follow the call of Kimmerer [10] for balanced respect of and support for EBK and TEK in our understanding of the world. Nutritional transition highlights the worldwide phenomenon of shifting patterns of human engagement in the food system. As societies change and food sourcing shifts from local territory to globalized supermarket shelves, they experience specific corresponding nutrition-related trauma of those arrangements [41]. It is true that dietary reliance on the most local of production can leave populations vulnerable to food insecurity and famine caused by routine seasonal and climate fluctuations, interferences with regular plant and animal reproduction, or social upheaval. But the dependency on the global supermarket coexists with conditions of uneven capital accumulation and speculation that have consistently led to polarized wealth and attendant poverty and food insecurity in rich and poor countries alike. Further, global food circulates with global capital and travels best in highly processed, shelf-stable forms that have been linked to the diminished consumption of local and traditional foods, loss of cultural food knowledge, less direct engagement with the food system, co-incidental sedentary lifestyles, and finally an epidemic of chronic disease. The agro-industrial pattern of food consumption and correlated nutritional deficiencies have been growing steadily in the Global North since the latter 19th century. In the Global South, this shift began more recently and is exploding in terms of both global market consumption, chronic disease prevalence, and loss of traditional food and nutritional knowledge in the attempt to “catch up” to the technology-driven (and unevenly distributed) wealth of the Global North.

In summary, we have compiled evidence demonstrating the potential antioxidant contribution of forageable foods to address oxidative stress and inflammation that are major players in the incidence of chronic disease. Where existing data allowed, we introduced a method to profile the theoretical synergistic potential of multiple antioxidants working together in select whole, unprocessed foods. We regret that the available nutrition data on forageables are incomplete, and we regard this as a violation of a public right to information about preventive health and well-being. We question why foods’ “nutritional content” is controlled by market participants who can afford expensive chemical analyses because they are “repaid” in profit. Public policy must prioritize, fund, and promote mechanisms to learn more about the nutritional benefits of wild, forageable foods, as well as expanded and non-commercialized food systems. Such efforts will support community and cultural food capital and help to realize food and nutrition rights associated with empowered community self-reliance.

## 6. Conclusions

This paper shows that forageable foods in a Central New York State city contain substantial amounts of five key antioxidant phytochemicals, indicating that “wild” foods have the potential to augment local dietary diversity and quality. These findings could be amplified in several ways. Most importantly, USDA and other researchers must expand and report micro- and macronutrient research on non- and less- commercialized forageable foods. Secondly, future studies should consider a review of a broader array of antioxidant phytochemicals, especially as nutritional biochemistry expands our understanding of how these bioactives function. Thirdly, comparable studies should be developed in diverse local ecosystems with alternate selections of forageable foods.

Public and private expansion of urban and rural food forests have implications for not only food and nutrient reserves but also other issues such as urban greening, climate change mitigation, riparian conservation, and sustainability. Support for food forests requires attention to safe and clean water and soils, including monitoring, pesticide-free maintenance, and public notice of risks and safety. Public and professional training is necessary. Nutritionists need expanded datasets for their community nutrition and research repertoire. Nutrition education can raise consumer awareness and encourage the incorporation of more non-commodified foods. Landscape professionals should reconsider design configurations that incorporate edibility. Food studies and food systems experts must magnify their vision of food access beyond commercial food systems, and even beyond backyard and allotment gardens. Public knowledge about and experience with safe and productive foraging needs to be advanced. And perhaps most of all, Indigenous knowledges about ethical foraging and the honorable harvest that sustain wild food ecologies must be respected.

## Figures and Tables

**Figure 1 nutrients-15-04630-f001:**
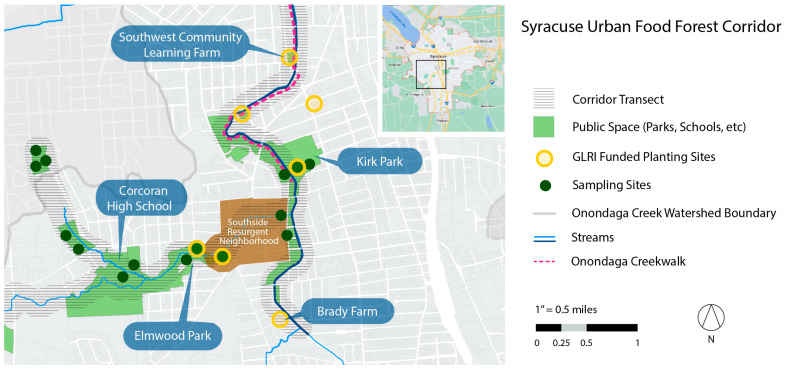
Syracuse Urban Food Forest Project (SUFFP), Riparian Forest Corridor along Furnace Brook and Onondaga Creek on Southside of Syracuse, NY.

**Figure 2 nutrients-15-04630-f002:**
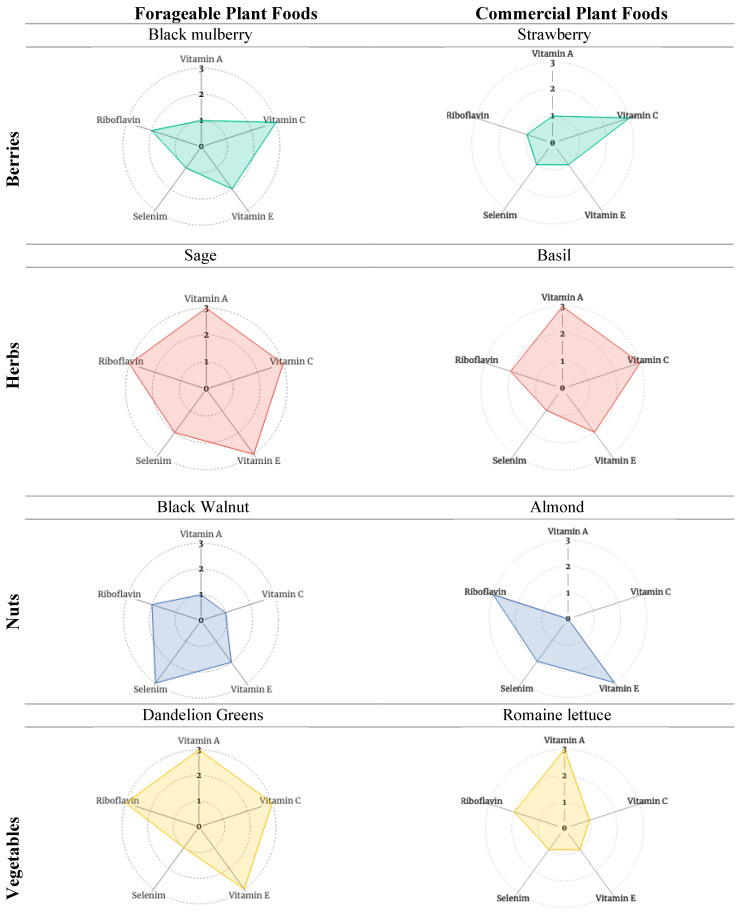
Radar charts of antioxidant scores of select forageable and commercial plant foods. Note: The individual antioxidants are shown on the “spoke” axes. The antioxidant “scores” (0–3), representative of the FDA’s classification of foods as no-concentration to high sources of an antioxidant, are shown on the “ring” axes.

**Figure 3 nutrients-15-04630-f003:**
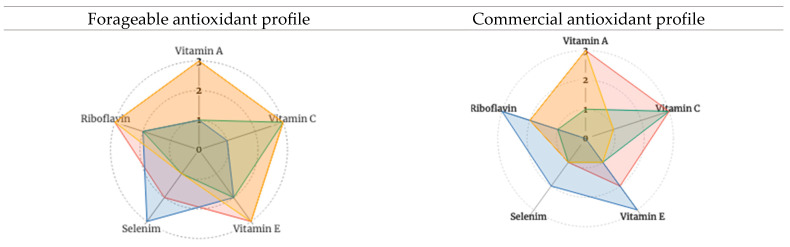
Combined antioxidant score profiles for select forageable and commercial plant foods Note: The individual antioxidants are shown on the “spoke” axes. The antioxidant “scores” (0–3), representative of the FDA’s classification of foods as no-concentration to high sources of an antioxidant, are shown on the “ring” axes.

**Table 1 nutrients-15-04630-t001:** Chronic Disease Rates and Food Insecurity.

Location		Cancer ^a^ (%)	Coronary Heart Disease (%)	Diabetes (%)	High Blood Pressure (%)	Kidney Disease (%)	Obesity (%)	Food Insecurity ^b^ LI = Low-Income LA = Low-Access ^c^
United States		7.5	8.2	12.7	37.1	3.5	36.0	
Onondaga County		7.4	6.3	9.7	30.7	2.9	31.9	----
City of Syracuse		5.5	6.4	10.9	30.9	3.2	35.9	----
Southside (mean of 4 Census Tracts)		5.2	8.9	19.0	44.1	5.2	49.1	----
	Census Tract 42	4.3	9.1	20.1	44.0	5.7	52.3	Not LI and/or LA
	Census Tract 53	6.4	11.0	20.3	46.7	5.8	47.9	LI and LA at 0.5 miles
	Census Tract 54	5.2	7.7	17.7	42.6	4.6	47.3	LI and LA at 0.5 miles
	Census Tract 58	5.0	7.6	17.9	43.2	4.6	48.9	LI and LA at 0.5 miles

Note. Rates are reported for adults aged 18 years or more [76]. ^a^ Excludes skin cancer. ^b^ [78]. ^c^ Low access is where a significant number of residents live more than the specified distance (e.g., 0.5 miles) from the nearest supermarket.

**Table 2 nutrients-15-04630-t002:** Food and Drug Administration Daily Values ^a^ and nutrient source thresholds ^b^.

	Assigned Score	Vitamin A (DV = 900 mcg)	Vitamin C (DV = 90 mg)	Vitamin E (DV = 15 mg)	Selenium (DV = 55 mcg)	Riboflavin (DV = 1.3 mg)
High source (≥20% DV)	3	≥180 mcg	≥18 mg	≥3 mg	≥11 mcg	≥0.26 mg
Medium source (20–5% DV)	2	45–180 mcg	4.5–18 mg	0.75– 3 mg	2.75–11mcg	0.065–0.26 mg
Low source (≤5% DV)	1	≤45 mcg	≤4.5 mg	≤0.75 mg	≤2.75 mcg	≤0.065 mg
No value	0	0	0	0	0	0

^a^ [82]. ^b^ [83].

**Table 3 nutrients-15-04630-t003:** Known antioxidant concentrations (high, medium, low) in edible, forageable plants ^a^.

* **Scientific Name** *	Common Name	Edible Part	Vitamin A (RAE mcg)	Vitamin C (mg)	Vitamin E (mg)	Se (mcg)	Riboflavin (mg)
**Fruits**							
*Amelanchier alnifolia*	Saskatoon serviceberry	raw fruit	10.91	3.55	1.12		3.54
*Asimina triloba*	pawpaw	raw fruit	8.6	18.3			0.09
*Celtis occidentalis*	hackberry	raw fruit				18.3	
*Chaenomeles speciosa*	flowering quince	raw fruit		72.15			
*Cornus kousa*	kousa dogwood	raw fruit		93			
*Diospyros virginiana*	persimmon ^b^	raw fruit		66			
*Malus* spp.	apple	raw fruit	3	4.6	0.18	0	0.026
*Malus* spp.	crabapple	raw fruit	2	8			0.02
*Morus alba*	white mulberry	raw fruit		15.2			0.088
*Morus nigra*	black mulberry	raw fruit	1	36.4	0.87	0.6	0.101
*Prunus americana*	American plum	raw fruit	173	10.3	0.53		0.042
*Prunus avium*	sweet cherry	raw fruit	3	7	0.07	0	0.033
*Pyrus* spp.	pear	raw fruit	1	4.3	0.12	0.1	0.026
*Ribes nigrum*	blackcurrant	raw fruit	12	181	1		0.05
*Ribes rubrum*	redcurrant	raw fruit	2	41	0.1	0.6	0.05
*Ribes uva-crispa*	gooseberry	raw fruit	15	27.7	0.37	0.6	0.03
*Rubus fruticosus*	blackberry	raw fruit	11	21	1.17	0.4	0.026
*Rubus idaeus*	red raspberry	raw fruit	2	26.4	0.57		0.08
*Sambucus canadensis*	black elderberry	raw fruit	30	36		0.06	0.06
*Vitis* spp.	grape	raw fruit	5	4	0.19	0.1	0.057
**Herbs**							
*Achillea millefolium*	yarrow ^c^	leaves			0.95		
*Mentha spicata*	spearmint	fresh leaves	203	13.3			0.175
*Salvia officinalis*	sage	ground herb	295	32.4	7.48	3.7	0.336
**Nuts**							
*Carya illinoinensis*	pecan	raw nut	3	1.1	1.4	3.8	0.13
*Corylus americana*	hazelnut	raw nut	1	6.3	15	2.4	0.113
*Juglans cinerea*	butternut	dried nut	6	3.2		17.2	0.148
*Juglans nigra*	black walnut	dried nut	2	1.7	2.08	17	0.13
*Juglans regia*	English walnut	dried nut	1	1.3	0.7	4.9	0.15
**Vegetables**							
*Alliaria petiolata*	garlic mustard	fresh leaves		26.1			
*Allium tricoccum*	ramps	whole plant		80			
*Chenopodium album*	lambsquarters	raw vegetable	580	80		0.9	0.44
*Matteuccia struthiopteris*	ostrich fern	raw fronds	181	26.6			0.21
*Nasturtium officinale*	watercress	fresh leaves	160	43	1	0.9	0.12
*Portulaca oleracea*	purslane	raw vegetable		21		0.9	0.112
*Taraxacum* spp.	dandelion	raw greens	508	35	3.44	0.5	0.26
*Typha latifolia*	broadleaf cattail	young shoots	1	0.7		0.6	0.025
*Vitis* spp.	grape	leaves	1380	11.1	2	0.9	0.354
**Other**							
*Acer saccharum*	sugar maple	sap	0	0	0	0.6	1.27

*Note.* All values are reported per 100 g of “edible part.” Color coding corresponds to the scores described in Table 2, where 3 is a “high source” (green), 2 is a “medium source” (peach), 1, “low source “and 0, “no value” are grouped together (blue). ^a^ USDA FoodData Central (FDC) uses unique ID numbers to identify food items [81]; ^b^ Persimmon is an example of a food for which the USDA provides values for macronutrients but limited micronutrients. ^c^ Yarrow (*Achillea millefolium*) samples were collected in Spain and Portugal.

**Table 4 nutrients-15-04630-t004:** Composite Antioxidant Scores of Forageable Plants.

* **Scientific Name** *	Common Name	Edible Part	Vit A Score	Vit C Score	Vit E Score	Se Score	Riboflavin Score	Composite Score
**Fruits**								
*Morus nigra*	black mulberry	raw fruit	1	3	2	1	2	9
*Rubus fruticosus*	blackberry	raw fruit	1	3	2	1	1	8
*Ribes uva*	gooseberry	raw fruit	1	3	1	1	1	7
*Ribes rubrum*	redcurrant	raw fruit	1	3	1	1	1	7
*Prunus avium*	sweet cherry	raw fruit	1	2	1	0	1	5
*Pyrus* spp.	pear	raw fruit	1	1	1	1	1	5
*Vitis* spp.	grape	raw fruit	1	1	1	1	1	5
*Malus* spp.	apple	raw fruit	1	1	1	0	1	4
**Herbs**								
*Salvia officinalis*	sage	ground herb	3	3	3	2	3	14
**Nuts**								
*Juglans nigra*	black walnut	dried nut	1	1	2	3	2	9
*Corylus americana*	hazelnut	raw nut	1	2	3	1	2	9
*Carya illinoinensis*	pecan	raw nut	1	1	2	2	2	8
*Juglans regia*	English walnut	dried nut	1	1	1	2	2	7
**Vegetables**								
*Taraxacum* spp.	dandelion	raw greens	3	3	3	1	3	13
*Vitis* spp.	grape	leaves	3	2	2	1	3	11
*Nasturtium officinale*	watercress	fresh leaves	2	3	2	1	2	10
**Other**								
*Acer saccharum*	sugar maple	sap	0	0	0	1	3	4

Note. Scores and color-coding correspond to the scores described in Table 2, where 3 is a “high source” (green), 2 is a “medium source” (peach), 1 is a “low source” and 0 is “no value” (blue).

**Table 5 nutrients-15-04630-t005:** Composite Antioxidant Scores for Forageable and Popular Commercial Plants.

* **Scientific Name** *	Common Name	Edible Part	Vit A Score	Vit C Score	Vit E Score	Se Score	Riboflavin Score	Composite Score
**Fruits**								
*Morus nigra*	black mulberry	raw fruit	1	3	2	1	2	9
*Fragaria* spp.	strawberry ^a^	raw fruit	1	3	1	1	1	7
			(1 mcg/100 g)	(58.8 mg/100 g)	(0.29 mg/100 g)	(0.4 mcg/100 g)	(0.022 mg/100 g)	
**Herbs**								
*Salvia officinalis*	sage	ground herb	3	3	3	2	3	14
*Ocimum basilicum*	basil ^b^	fresh herb	3	3	2	1	2	11
			(264 mcg/100 g)	(18 mg/100 g)	(0.8 mg/100 g)	(0.3 mcg/100 g)	(0.076 mg/100 g)	
**Nuts**								
*Juglans nigra*	black walnut	dried nut	1	1	2	3	2	9
*Prunus dulcis*	almond ^c^	raw nut	0	0	3	2	3	8
			(0 mcg/100 g)	(0 mg/100 g)	(25.6 mg/100 g)	(4.1 mcg/100 g)	(1.14 mg/100 g)	
**Vegetables**								
*Taraxacum* spp.	dandelion	raw greens	3	3	3	1	3	13
*Lactuca sativa*	romaine lettuce ^d^	raw leaves	3	1	1	1	2	8
			(436 mcg/100 g)	(4 mg/100 g)	(0.13 mg/100 g)	(0.4 mcg/100 g)	(0.067 mg/100 g)	

^a^ FDC 167762. ^b^ FDC 172232. ^c^ FDC 170567. ^d^ FDC 169247.

## Data Availability

For information on data used to support this study, contact Anne C Bellows at acbellow@syr.edu.

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
