# Peer review of "Foraging Wild Edibles: Dietary Diversity in Expanded Food Systems"

_nutrients, 2023, doi:10.3390/nu15214630_

Round 1
Reviewer 1 Report
Thank you very much for the opportunity to read the text.
The paper presented is fairly written and meets the requirements for scientific papers. It contains necessary elements such as limitations and future recommendations. A tabular appendix is also included.
The topics are quite interesting.
I would suggest the authors only shorten the title, as it is too long.
I also don't see an isolated hypothesis or research questions in the paper. Please include them in the section with the aim of the paper.
Reviewer 2 Report
This research examined the nutrient/phytochemical content of foraging wild edibles and compared these contents with popular commercially available comparison foods. This is an interesting read, but there is much repeat information and the text can be condensed considerably. The ‘Recommendations’ section is nicely stated.
Although it is worthy to consider the nutrient contribution of edibles, since these plants are not cultivated for consumption, there is the concern of toxicity which is not addressed in the text by the authors. As example, a recent story in the Tri-City Herald warned: “resist the urge to eat anything you pick while out in the Washington wilderness, unless you are 100% certain what it is” since there are poisonous look-alikes. Furthermore, there is a possibility that plants have been sprayed with weed killer, pesticides or other harmful sprays. Moreover, while parts of edibles are safe to eat, the bark, stems and roots may be poisonous. The Cleveland Clinic states: “While some [wild plants] are edible, the risk of picking a poisonous one is too high”. These topics deserve coverage in this article.
Is it possible that anti-nutritional factors, such as saponins, tannins, phytic acid, gossypol, lectins, protease inhibitors, amylase inhibitor, and goitrogens, are more prevalent in wild edibles as these plants have not been cultivated for consumption? Different technological processing methods (as well as traditional, cultural methods) are currently used (e.g, milling, debranning, roasting, germination and fermentation) to reduce these anti-nutritional components in foods to enhance nutrient bioavailability. This is another consideration that should covered in this article.
Abstract: suggestion – replace ‘lacunae’ with gap
Figures 2 and 3 can be combined (as was done in Figure 4) to focus the comparison.
Reviewer 3 Report
The manuscript focuses on very interesting subject in the field of public health nutrition, i.e., the estimation of antioxidant score of foraging wild edibles to investigate the nutritional value of increasing diet diversity.
The manuscript could benefit of additional revision according to the following suggestions:
(1) Selection of keywords: the keywords included in the manuscript may reduce the probability of identification and citation of the paper, since comprise complex terms; my suggestion is to focus on: diet diversity, bioactive compounds, antioxidants, urban gardens, nutritional composition.
(2) Reduction of bias in approaching food systems: authors should revise the approach of food systems, since the evidence on globalization of diets is contradictory. Whilst certain studies show that globalization increase food diversity (e.g., https://doi.org/10.1016/j.crfs.2023.100517 and https://doi.org/10.1111/twec.12260 and https://doi.org/10.1371/journal.pone.0175554 and https://doi.org/10.1186/s12992-019-0457-y), others show the contrary (https://doi.org/10.1016/j.gfs.2021.100490 and https://doi.org/10.1016/j.socscimed.2015.03.030 and https://doi.org/10.1016/j.foodpol.2016.10.001). Thus, the literature review and the approach on food systems transformation should include the dilemma involving globalization of food systems, and that the use of local produce and "wild food" should be part of food consumption choices to ensure nutritional balance of diets.
(3) Reduction of the bias towards "traditional food systems": food consumption includes foods obtained from commerce between nations in our diets since Ancient times, considering that only foraging usually does not ensure proper calorie intake, and thus population survival/growth. The authors emphasize the potential "war" between the traditional and the commercial food production, although the food systems should comprise various production, processing and distribution methods to benefit the population, and guarantee better nutritional and health status of individuals. In fact, the authors indicate that industrialized economies "are also associated with habitat destruction, loss of land tenure, and rural out-migration that undermines traditional food activities like foraging"; however, there are evidence on the contrary. In fact, some large-scale food production enterprises adopt rules of conservation and use of pesticides more closely than small-scale businesses in certain countries (e.g., https://doi.org/10.1098/rstb.2012.0378).
(4) Improvement of linkage between arguments presented in the Introduction: the fourth paragraph (page 2, lines 52-68) needs to be linked with the previous paragraph, since it is disconnected from the text flow. The previous paragraphs lack mentions to phytochemicals, their importance to human nutrition/health, or their consumption level on population diet. The link could be improved through additional literature review, considering that the subject comprises the core of the manuscript (e.g., https://doi.org/10.2174/1381612823666170317122913 and https://doi.org/10.3390/plants12132414 and https://doi.org/10.1073/pnas.17091941 and https://doi.org/10.1017/S0007114518001381 and https://doi.org/10.1007/s00394-015-0942-x)
(5) Substitution of the term "non-communicable chronic degenerative diseases: usually, the term "non-communicable chronic diseases (NCD)" is sufficient in studies of the field of knowledge.
(6) Revision of the approach of "dual burden of disease": the dual burden of disease phenomenon has been debated in the last decades of the 20th century, being recently replaced by the problem of the triple burden of malnutrition; thus, it is recommended that authors change the focus of the argument presented in the manuscript (page 3, lines 135-148).
(7) Inclusion of additional limitations of the study: authors should acknowledge concerns on the contamination of urban areas, including land and water pollution with heavy metals, microplastics, and others (e.g., https://doi.org/10.1016/j.swaqe.2015.04.002 and https://doi.org/10.1007/978-4-431-78399-2_7 and https://doi.org/10.1071/EN14167 and https://doi.org/10.1007/s10661-019-7947-5 and https://doi.org/10.3390/ijerph18105164 and https://doi.org/10.1016/j.envint.2021.106582).
(8) Change the first paragraph of the subsection "Antioxidant Phytochemical Data Collection": the first paragraph (page 8, lines 362-366) comprises the justification of the choice of foods with data on "vitamins A, C, and E, selenium, and/or riboflavin content", which is mentioned in the previous subsection (page 8, line 353).
(9) Inclusion of limitations regarding the strategy for selection of foods for the analysis: the strategy based on the selection of foods with data on certain nutrients probably resulted in exclusion of the most traditional and indigenous plants, since government datasets of nutrition composition usually focus on the foods consumed by wide rage of individuals and population groups.
(10) Revision of manuscript formatting: there are problems in terms of font size (e.g., page 16, lines 594-596), use of bold letters (e.g., page 20, line 814), or capitalization of words in tables (e.g., Table 4, column "Common names"), among others.
Round 2
Reviewer 2 Report
Thank you for addressing my concerns. Good luck with the paper.
Author Response
Thank you for your time & feedback!